# Identification of the Knowledge Structure of Cancer Survivors’ Return to Work and Quality of Life: A Text Network Analysis

**DOI:** 10.3390/ijerph17249368

**Published:** 2020-12-14

**Authors:** Kisook Kim, Ki-Seong Lee

**Affiliations:** 1Department of Nursing, Chung-Ang University, Seoul 06974, Korea; kiskim@cau.ac.kr; 2Da Vinci College of General Education, Chung-Ang University, Seoul 06974, Korea

**Keywords:** cancer survivors, return to work, quality of life, research trend, text network analysis

## Abstract

This study aimed to understand the trends in research on the quality of life of returning to work (RTW) cancer survivors using text network analysis. Titles and abstracts of each article were examined to extract terms, including “cancer survivors”, “return to work”, and “quality of life”, which were found in 219 articles published between 1990 and June 2020. Python and Gephi software were used to analyze the data and visualize the networks. Keyword ranking was based on the frequency, degree centrality, and betweenness centrality. The keywords commonly ranked at the top included “breast”, “patients”, “rehabilitation”, “intervention”, “treatment”, and “employment”. Clustering results by grouping nodes with high relevance in the network led to four clusters: “participants and method”, “type of research and variables”, “RTW and education in adolescent and young adult cancer survivors”, and “rehabilitation program”. This study provided a visualized overview of the research on cancer survivors’ RTW and quality of life. These findings contribute to the understanding of the flow of the knowledge structure of the existing research and suggest directions for future research.

## 1. Introduction

The number of cancer survivors is rapidly increasing due to early screening and improved medical technology. As the cancer survival rates increase, optimizing the healthcare provision and long term outcomes for survivors becomes increasingly imperative [1].

For cancer survivors, returning to work (RTW) signifies more than economic power. It represents returning to daily life, from which they were temporarily excluded because of the cancer diagnosis and treatment, and a significant factor affecting their quality of life through the restoration of interpersonal relationships and social status [2,3,4]. Cancer survivors resume their daily life, including RTW, during or after treatment; however, the RTW rate varies from 30.5% to 71.5% [5,6]. Many cancer survivors have difficulty RTW. Moreover, there are cases of unemployment or job turnover due to the side effects of cancer treatment and the social stigma associated with cancer patients [5]. The unemployment rate among cancer survivors differs based on their diagnosis; nonetheless, cancer survivors have a 1.4 times higher unemployment rate than healthy controls [7].

After their treatment, RTW cancer survivors experience side effects, such as fatigue, pain, restriction of physical function and activity, anxiety, and depression, which lead to difficulties in re-adaptation to work-life balance, job performance, and interpersonal relationships [3,8]. These complex factors increase the job stress of cancer survivors, hinder a successful return to work, and result in job turnover, leading to a decrease in their overall quality of life, which, in turn, acts as a restricting factor for cancer survivors’ multi-faceted difficulties and successful RTW [4,9]. Research in various academic fields is required to explore the comprehensive range of factors related to the successful RTW of cancer survivors. Additionally, it is necessary to prioritize the research status and structural relationship of cancer survivors RTW and their quality of life. Thus, several factors are associated with RTW and quality of life based on specific cancer diagnosis [10,11,12], employment type [13], and prevalence [6,12]. A systematic review of interventions to enhance the RTW [14] of cancer survivors has been conducted. Nevertheless, there is limited research focusing on constructing an efficient program for improving the quality of life of RTW cancer survivors based on a macroscopic view of the knowledge structure of research trends. The methods for comprehensive literature analysis have time, labor, and accuracy limitations as they comprise a wide range of research subjects and a vast literature. 

Big data processing, or new data analysis, includes both multivariate statistics, such as logistic regression, and modern methods, such as social network analysis, natural language processing, data mining, and machine learning techniques [15]. Among those, social network analysis has been used as a method for knowledge discovery to identify patterns and trends in various health disciplines [16,17,18]; that is, not only the frequency of appearance of words related to a specific subject, their ranking, and what information they have, but also the relationship between the words, the logic by which they are positioned according to context, and their overall structure. As a result, it is possible to understand the subject thoroughly and visually around the texts surrounding the subject [19,20]. Because the text network analysis is a type of quantitative content analysis, it is possible to identify core topics, interpret potential contextual meanings, and visualize knowledge structures based on the relationship between word co-occurrences in large-scale text [21].

Knowledge structure is defined as a contextual structure between knowledge structure research concepts that show the boundaries or development process of knowledge [22,23]. Major research concepts discovered through scientific methods by quantifying words and their arrangements used in the research literature are considered to be the theoretical basis of the academic field as the core topics of the knowledge structure [21]. Thus, to explore the knowledge structure of a specific academic field, it is necessary to quantify the major research concepts that interest many researchers, recognize the contextual structure of the main concepts, and identify which research concepts are discussed more often [22,23,24]. 

Therefore, in this study, the knowledge structure of research related to the quality of life of RTW cancer survivors was identified and the association between topics was visualized using text network analysis.

## 2. Materials and Methods 

### 2.1. Design

This quantitative content analysis study applied the text network analysis method to previously published literature on the quality of life of RTW cancer survivors.

### 2.2. Research Procedure and Method

Data collection was conducted through literature search and text collection. The titles and abstracts from the literature on RTW and the quality of life of cancer survivors was set as the data range. For the title analysis, we constructed a semantic network based on the simultaneous appearance of keywords and analyzed the attributes observed in the network. For the abstracts, we introduced a hierarchical topic model [25] and analyzed the main topics and their relationships. The research procedure was as follows: (1) literature data collection, (2) keyword extraction and preprocessing, (3) title semantic network creation and analysis, and (4) hierarchical topic model generation and analysis of abstracts (Figure 1).

### 2.3. Search and Collection of Articles

Research literature was collected from PubMed, Embase, Cochrane, the Cumulative Index to Nursing and Allied Health Literature, Web of Science, and Scopus using EndNote (Clarivate Analytics, Boston, MA, USA). Articles containing “cancer survivors”, “return to work”, and “quality of life” in the title or abstract were selected and vocabulary combinations with similar meanings were also considered (e.g., “back to work” or “job re-entry”). After excluding duplicates, 219 articles published between 1990 and June 2020 were collected. For text mining, titles and abstracts were organized into one row and saved as an Excel file. Articles without an abstract were also excluded from the collection.

### 2.4. Keyword Extraction and Preprocessing 

To extract keywords, punctuation and numbers in the title and abstract were removed. Afterward, the parts of speech of all the words were identified and only nouns, verbs, and adjectives were kept and lemmatization was performed to convert them to their base form. Finally, by removing stopwords and keywords directly used in the literature search such as “cancer”, “survive”, and “work”, the titles and abstracts were reconstructed with only meaningful words. Furthermore, for a rough analysis, the frequency of words and n-grams (n consecutive word combinations) were measured [26]. For these preprocessing, we developed a tool that uses the Natural Language Toolkit library (https://www.nltk.org/; Free and open-source software) in the Jupyter notebook (https://jupyter.org/; Free and open-source software) environment based on the python programming language. 

### 2.5. Semantic Network Analysis for Title

The semantic network is a graph that expresses words and their relationship as nodes and edges, respectively [27]. To establish word associations, this study defined word pairs appearing together in a paper’s title as related to each other. Each node (word) is connected to other nodes in the network and the number of connections is called “degree” of that node. The degree value of a node indicates its importance and degree centrality quantifies how central each node is as a value between 0 and 1. Moreover, betweenness centrality is a value between 0 and 1 that indicates how much each node acts as a bridge for the connections between other nodes.

Besides these numerical measurements, network analysis was performed using a clustering technique for grouping nodes with high relevance on a network. We applied Blondel’s model [28], based on modularity, which organizes a network so that it has high intraconnection and low interconnection. This technique makes it possible to identify which words are highly related to each other within groups. Clustering and visualization were performed using a sub-network composed of selected main keywords only because the original semantic network was too large to present in its entirety. A Python library, NetworkX (https://networkx.org/; Free and open-source software), was used for network creation and analysis and a Gephi (https://gephi.org/; Free and open-source software) tool was used for clustering and visualization.

### 2.6. Hierarchical Topic Analysis for Abstract

Topic analysis is a technique based on unsupervised machine learning that infers what topic is embedded in a large number of text documents. A commonly used topic analysis technique is Latent Dirichlet Allocation (LDA) [29], which represents all topics in a flat relationship. This study applied the hierarchical LDA (hLDA) [25], which re-organizes topics into a vertical hierarchical structure to interpret large documents more easily. The hLDA uses the nested Chinese Restaurant Process [30] and LDA and is explained by the following analogy:There are Chinese restaurant chains organized into a tree structure.A guest eats in one restaurant and then moves to the next restaurant in the subchain.There are many tables and seats in each restaurant and guests choose seats based on the popularity of the table.The popularity of the table is proportional to the number of seated guests.What food is placed on the table is determined by contacting the upper chain restaurant.

Following these assumptions, each time a new guest comes, the type of food and the popularity of the table converge on a certain value. In the hLDA, the food served by the Chinese restaurants at each level represents a topic and the guests represent the document. As guests visit the Chinese restaurants at different levels, the n-layer foods they eat correspond to the n-layer topics of the document. This allows us to identify hierarchical topic structures and to which topics each document belongs. The hLDA requires user input on how many layers a topic will be composed of and in this study, several models were created with three and four layers. Among the generated topic models, one analytically representative model was selected and a detailed analysis was performed. For topic extraction tools, genism (https://radimrehurek.com/gensim/; Free and open-source software) and hLDA (https://github.com/joewandy/hlda/; Free and open-source software) libraries were used.

## 3. Results

### 3.1. Core Keywords that Emerged from the Research Titles

Table 1 presents the top 20 core keywords by frequency, edge, degree centrality, and betweenness centrality indices calculated from the main words extracted from the studies on the quality of life among RTW cancer survivors. There were 587 unique words extracted from the selected 219 research titles. Therefore, the title semantic network had 587 nodes.

Regarding frequency, “breast”, “patients”, “rehabilitation”, “intervention”, and “treatment” were ranked at the top. The edges and degree centrality were also high in the order of “breast”, “patients”, “intervention”, “rehabilitation”, “employment”, and “treatment”. The frequency, edge, and degree centrality were shown to have similar rankings for most of the core keywords. There were minor differences in rankings; however, after the seventh place, the keywords “development”, “follow”, “experience”, and “survivorship” were ranked relatively high in betweenness centrality compared to frequency and edge.

### 3.2. Semantic Network Analysis

There were 4346 word pairs, resulting in a semantic network with 587 nodes and 4346 edges. The average degree value of the network was 17.8 and we selected important nodes with a degree of 30 or higher, which represented 10% of all nodes. Figure 2 shows the semantic network diagram for the top 10% based on degree centrality.

It was divided into four clusters and classified by font size and color according to connectivity. The keyword “breast” was clustered with keywords such as “treatment”, “survivorship”, “diagnosis”, “employment”, “association”, “impact”, “symptom”, “change”, and “prospective” and was labeled as “participants and method”. The keyword “patient” was clustered with keywords such as “intervention”, “psychosocial”, “control”, “development”, “systematic”, “protocol”, “trial”, “support”, and “randomize” and was labeled as “type of research and variables”. Additionally, the keywords “adult”, “young”, “stem”, “cell”, “transplantation”, “education”, “early”, and “experience” formed another cluster labeled as “RTW and education in adolescent and young adult cancer survivors”. The keywords “rehabilitation”, “occupational”, “program”, and “pilot” formed another cluster labeled as “rehabilitation program”.

### 3.3. Hierarchical Topic Analysis for Abstracts

Words dealing with the same subject were assumed to appear together frequently and a topic could be inferred by identifying a set of these words. About 29,307 words appeared in the collected abstracts of 219 papers, of which 3901 were unique. Among the generated hLDA topic models, researchers selected one analytically representative model with three layers (Table 2). 

From the hierarchical topic analysis on the three layers, two topics were derived from level 1 and 11 topics were derived from level 2.

The 11 identified topics were classified as follows: rehabilitation intervention (intervention, rehabilitation, trial, exercise, program), employment and symptom (employment, diagnosis, status, symptom, confidence interval), diagnosis and job status (month, diagnosis, employment, status, job), psychosocial factors (pain, fear of cancer recurrence, lymphedema, barrier, surgeon), health-related quality of life by group (health-related, confidence interval, health-related quality of life, oral, group), physical exercise (exercise, lung, physical, patient, improve), education of young patient (adult, adolescent and young adults, educational, service, young), job type and quality of working (self-employed, item, the quality of working life questionnaire for cancer survivors, module, job), analytic method (literature, search, criterion, systematic, productivity), cost (engagement, consequence, cost, stakeholder, provide), and intervention for hematopoietic stem cell transplant (hematopoietic stem cell transplant, yoga, cognitive, transplantation, standard care).

## 4. Discussion

This study analyzed the literature on the quality of life of RTW cancer survivors to understand the flow of knowledge structure of the existing research and suggest directions for future research. The study found that the core keywords of the quality of life of RTW cancer survivors research included “breast”, “patients”, “rehabilitation”, “intervention”, “treatment”, and “employment”. The keywords with high centrality were regarded as core keywords [31]; most of these high centrality keywords were also high in frequency.

“Breast” and “patient” were ranked high in frequency, degree centrality, and betweenness centrality. Breast cancer is a carcinoma with high prevalence in women. Since the development of diagnostic and treatment methods has led to a high survival rate and long duration of survival, breast cancer is being studied in many areas. It is consistent with the previous study [22], which examined the knowledge structure of cancer survivors and showed that breast neoplasm had a higher frequency after quality of life. 

Individual perception of discrimination and lack of support from employers and colleagues can negatively affect successful workplace participation and is more serious for female cancer survivors [32,33]. Women’s resumption of work is negatively affected by various human resource factors [34]. Therefore, it is necessary to examine more diverse strategies and conduct intervention development studies for RTW in breast cancer patients.

RTW has been treated as a concept related to social, vocational, and physical rehabilitation for the workplace adaptation of cancer survivors, indicating that many studies are related to intervention performance. Successful vocational rehabilitation has a major impact on RTW skills and helps maintain the quality of working life of cancer survivors [35]. Additionally, cancer survivors are affected by work capacity due to neuropathy, fatigue, and chemo or radiation therapy [10,36]. Therefore, various studies have been conducted to improve their quality of life according to the ongoing treatment. 

In this study, the degree centrality of the title keywords was visualized as a sociogram. In the sociogram, the size of the node represented the grade of degree centrality and the thickness of the node represented the connection strength, that is, the frequency of simultaneous occurrence.

Keywords with high degree centrality are connected to many other keywords and are located at the center of the network and thus, represent an important core topic. In this study, most of the high ranking keywords in frequency, degree centrality, and betweenness centrality appeared prominently in the sociogram. However, keywords such as “development”, “experience”, and “survivorship”, which were not ranked high for edge, frequency, and degree centrality, were relevant as mediating roles, that is, in betweenness centrality. 

This is important because keywords with high betweenness centrality act as a mediator between other keywords and function as a bridge to expand from one topic to another [37,38], even though the frequency and edge are not relatively high. It is believed that a keyword plays a significant role in the knowledge structure.

Clustering results by grouping nodes with high relevance in the network. Based on connectivity, the keywords were divided into four clusters. Research on the quality of life of RTW cancer survivors was divided into the following groups: “participants and method”, “type of research and variables”, “return to work as education in adolescent and young adult cancer survivors”, and “rehabilitation program”. It implies that research is being conducted on the successful RTW of cancer survivors by applying various research methods. Moreover, previous studies have primarily used randomized trials, including physical exercise and psychosocial support of chemotherapy subjects, tailored intervention development, and systematic reviews to evaluate the effectiveness of these interventions.

In particular, intervention studies based on occupational rehabilitation and experience and education issues of RTW, such as education in adolescent and young adult cancer survivors, were considered important and found to be highly connected by the current study.

Although various intervention studies have been conducted, the effect of psycho-educational interventions without vocational rehabilitation is unclear. Therefore, a vocational rehabilitation program that includes work coordination and supervisor-centered vocational components rather than a patient-centered occupational environment program should be provided [14].

This was also studied as an important keyword in the topic analysis of the abstracts. The contents on job status, type, and quality of life were presented as a topic and issues related to cost and stem cell transplantation in adolescents and young adults were grouped as another topic. Compared to salaried workers, self-employed cancer survivors may have more difficulty in performing their jobs after RTW because they have less social support at work and less legal support from related laws and public health insurance [13]. Therefore, further research on the RTW of cancer survivors according to type of work and employment is needed. Many stem cell transplant cancer survivors complain of impairment in daily life functioning due to cognitive impairment, which is associated with younger age and reduced health-related quality of life [39]. In particular, long survival rates and cognitive impairment in cancer survivors are related to education, which in turn is related to finding a quality job, which is an important issue that needs to be addressed. 

RTW is regarded as a marker of complete recovery and restored normality [9,14]. It is often characterized as a complex and prolonged trajectory [40]. While several pieces of evidence indicate that quality work is beneficial for the physical and mental health of cancer survivors, unemployment and long term absence of illness have detrimental effects [14,41].

Furthermore, unsuccessful RTW has a significant impact on the health care system and on insurance, of which direct or indirect social costs are paid by patients and their families, employers, and society [42]. Therefore, the RTW of cancer patients is an issue that should be dealt with not only individually, but also socially. Based on the trends found in the present study, future research should consider various variables for RTW and quality of life improvement.

The strength of this study lies in the text network analysis that enabled us to identify the knowledge structure and topics related to RTW cancer survivors and quality of life research effectively, objectively, and comprehensively, thus providing the basis for the continuous improvement of RTW intervention research. However, this study also has certain limitations. Because the extracted text was collected only from the titles and abstracts of published papers, the ultimate purpose and meaning of each study may have been excluded. There may have been an impact of the time-dependent nature of the study that was not reflected in the present analysis.

## 5. Conclusions

Through network analysis and clustering, keywords were divided into four knowledge structures. Similar results were confirmed using hierarchical topic analysis on the abstracts. The study divided the research subjects into four clusters and it found that studies have been conducted using a psychosocial approach, young cancer survivors, breast cancer patients, rehabilitation, and interventions, which appeared as keywords. The subject matter of the study conducted through this knowledge structure can be identified, which helped reveal the research trend and direction of the knowledge structure related to the successful RTW and work-related quality of life of cancer survivors.

This study showed the visualized trends of the knowledge structure and research direction based on the previously published literature. However, it was difficult to confirm the knowledge body about direct demands such as disturbance and facilitation factors associated with RTW experienced by cancer survivors. Therefore, it will be necessary to develop RTW interventions that reflect the needs of cancer survivors and trends of the times through network analysis based on the direct responses of cancer survivors on online big data, such as online news comments and social network services.

In addition, through this study, trends and limitations in the diversity of related studies and the importance of expanding research on various intervention programs and accumulating evidence of their impact on the quality of life of RTW cancer survivors were identified.

## Figures and Tables

**Figure 1 ijerph-17-09368-f001:**
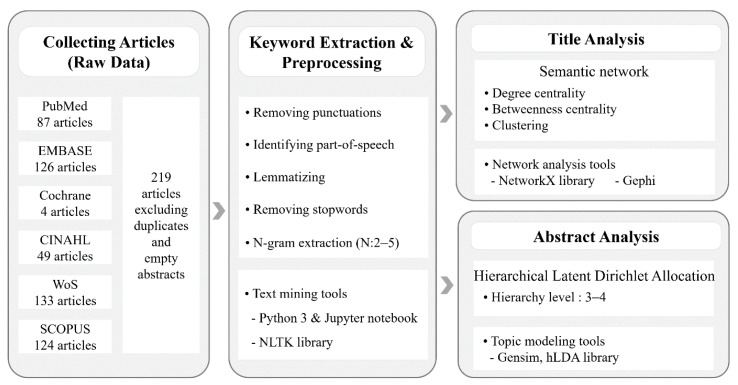
Flow diagram for the text analysis process.

**Figure 2 ijerph-17-09368-f002:**
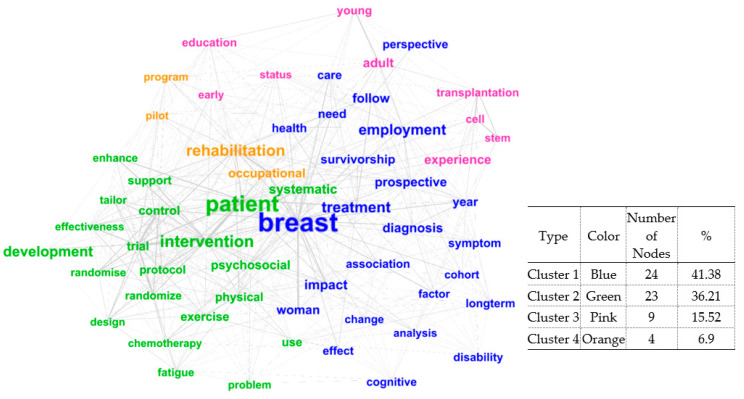
Clustering and semantic network diagram for keywords.

**Table 1 ijerph-17-09368-t001:** Top 20 core keywords in the research titles.

Rank	Keyword	Frequency	Keyword	Edges	Keyword	Degree Centrality	Keyword	Betweenness Centrality
1	breast	52	breast	182	breast	0.305	breast	0.188
2	patients	32	patient	164	patient	0.271	patient	0.138
3	rehabilitation	31	intervention	138	intervention	0.194	intervention	0.067
4	intervention	27	rehabilitation	125	rehabilitation	0.170	rehabilitation	0.060
5	treatment	20	employment	117	employment	0.149	treatment	0.052
6	employment	20	treatment	103	treatment	0.148	employment	0.049
7	systematic	16	trial	90	trial	0.132	development (38)	0.045
8	trial	15	impact	89	impact	0.130	impact	0.032
9	occupational	15	occupational	81	occupational	0.120	systematic	0.030
10	impact	15	control	81	control	0.118	diagnosis	0.027
11	year	14	woman	73	woman	0.110	adult	0.026
12	prospective	13	prospective	71	prospective	0.106	follow (22)	0.026
13	diagnosis	13	adult	67	adult	0.103	prospective	0.026
14	physical	12	support	64	support	0.103	experience (33)	0.025
15	woman	11	systematic	63	systematic	0.103	woman	0.023
16	follow	11	physical	63	physical	0.099	psychosocial	0.022
17	control	11	randomize	63	randomize	0.099	occupational	0.022
18	adult	11	diagnosis	61	diagnosis	0.098	survivorship (27)	0.020
19	psychosocial	10	health	61	health	0.098	year	0.019
20	need/exercise	10	psychosocial/year	60	psychosocial/year	0.098	control	0.019

Note: The numbers in parentheses are the corresponding rankings for edge and degree centrality.

**Table 2 ijerph-17-09368-t002:** Abstract topic analysis.

Level 0	n ^1^	Level 1	n	Level 2	n
treatment, patient, quality, life, year,	219	patient, intervention, breast, support, care	103	intervention, rehabilitation, trial, exercise, program	42
				employment, diagnosis, status, symptom, ci ^2^	19
				month, diagnosis, employment, status, job	26
				pain, fcr ^3^, lymphedema, barrier, surgeon	16
		need, support, breast, intervention, health	116	hr ^4^, ci, hrqol ^5^, oral, group	22
				exercise, lung, physical, patient, improve	12
				adult, ayas ^6^, educational, service, young	15
				self-employed, item, qwlqcs ^7^, module, job	13
				literature, search, criterion, systematic, productivity	22
				engagement, consequence, cost, stakeholder, provide	18
				hsct ^8^, yoga, cognitive, transplantation, sc ^9^	14

^1^ n: documents; ^2^ ci: confidence interval; ^3^ fcr: fear of cancer recurrence; ^4^ hr: health-related; ^5^ hrqol: health-related quality of life; ^6^ ayas: adolescent and young adults; ^7^ qwlqcs: the quality of working life questionnaire for cancer survivors; ^8^ hsct: hematopoietic stem cell transplant; ^9^ sc: standard care.

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
