# Peer review of "Identification of the Knowledge Structure of Cancer Survivors’ Return to Work and Quality of Life: A Text Network Analysis"

_ijerph, 2020, doi:10.3390/ijerph17249368_

Round 1
Reviewer 1 Report
The present ms provides a bibliometric analysis of quality of life to return to work.
My comments are as follows:
1. I was unsure about the primary goal of the paper. There is some bibliometric analysis of titles and abstracts. But how do these methods help to systemize the literature? Maybe, it would be sufficient to extend the introduction substantially and provide a more thorough discussion of the results.
2. I also struggled with the term “knowledge structure“. Is it meant as a technical term? If yes, there is no introduction in the ms about this term. If no, I do not think that the bibliometric analysis findings can be seen as generating “knowledge structure“. By reading the conclusion, it seems that the term is meant to be topic clusters?
3. A lot of concepts and software applications are introduced in the ms without providing references for it. For example, “text network analysis“ (67), „“hierarchical topic model“ (74), “n-grams“ (94), “NLTK library“ (95), to mention a few.
4. It would be helpful to describe better the technique of hierarchical latent Dirichlet allocation for readers unaware of these kinds of models. To me, it is only important to reflect on the main idea and model equations of this model.
Reviewer 2 Report
In this work, the knowledge structure of research related to the quality of life of RTW cancer survivors is identified and the association between topics is visualized using the text network analysis. Experimental results prove the effectiveness of the proposed method and it has a good application prospect. However, I believe that it should be carefully revised before publication. First, the sections of Abstract and Conclusions should be rewritten to highlight the main contributions of this work. In this situation, it is difficult for me to really understand the purpose and novelty of this work. Second, the section of Materials and Methods is not very clearly to present the proposed method. For example, some details of this work should be carefully addressed. Third, is it necessary to include the title of Section 4.1.
Round 2
Reviewer 1 Report
I am satisfied with the revised ms and the responses to my comments.
Reviewer 2 Report
Compared with the original version, the structure of this version of the paper is more reasonable. I am very satisfied with the author’s careful revision.